# The Prevalence of Viral Pathogens among Bats in Kazakhstan

**DOI:** 10.3390/v14122743

**Published:** 2022-12-09

**Authors:** Adilbay I. Karagulov, Takhmina U. Argimbayeva, Zamira D. Omarova, Ali B. Tulendibayev, Laura Zh. Dushayeva, Marina A. Svotina, Aibarys M. Melisbek, Nurdos A. Aubakir, Sabyrkhan M. Barmak, Kulyaisan T. Sultankulova, Dana A. Alibekova, Tanat T. Yermekbai, Askar M. Nametov, Dmitry A. Lozovoy, Khairulla B. Abeuov, Mukhit B. Orynbayev

**Affiliations:** 1Non Profit Joint Stock Company Zhangir Khan West Kazakhstan Agrarian Technical University, Institute of veterinary medicine and animal husbandry, Uralsk, st. Zhangir Khan 51, 090009, Kazakhstan; 2RSE Scientific Research Institute of Biological Safety Problems, SC MES RK, town. Gvardeisky, Korday district, Jambyl region, 080409, Kazakhstan; 3al-Farabi Kazakh National University, Faculty of Biology and Biotechnology, Republic of Kazakhstan, Almaty, 71 al-Farabi Ave., 050040, Kazakhstan; 4Russian Academy of Agro-Industrial Complex Staffing, Moscow, 600901, Russia

**Keywords:** bat, alphacoronaviruses, rabies virus, Kazakhstan, viral diversity, phylogenetic analysis

## Abstract

Bats carry thousands of viruses from 28 different families. To determine the presence of various pathogens in bat populations in Kazakhstan, 1149 samples (393 oropharyngeal swabs, 349 brain samples, 407 guano) were collected. The samples were collected from four species of bats (*Vespertilio murinus*, *Nyctalus noctula*, *Myotis blythii*, *Eptesicus serotinus*) in nine regions. The Coronavirus RNA was found in 38 (4.75%) samples, and the rabies virus in 27 (7.74%) samples from bats. Coronaviruses and the rabies virus were found in bats in six out of nine studied areas. The RNAs of SARS-CoV-2, MERS, TBE, CCHF, WNF, influenza A viruses were not detected in the bat samples. The phylogeny of the RdRp gene of 12 samples made it possible to classify them as alphacoronaviruses and divide them into two groups. The main group (*n* = 11) was closely related to bat coronaviruses from Ghana, Zimbabwe and Kenya. The second group (*n* = 1) was closely related to viruses previously isolated in the south of Kazakhstan. The phylogeny of the N gene sequence from a bat from west Kazakhstan revealed its close relationship with isolates from the Cosmopolitan group of rabies viruses (Central Asia). These results highlight the need for a continuous monitoring of volatile populations to improve the surveillance and detection of infectious diseases.

## 1. Introduction

The second largest group of mammals is the *Chiroptera* order. Bats occur in all continents except Antarctica and account for about 22% of all named mammal species. They are the perfect reservoir for viruses, and their unique ability to fly contributes to the spread of viruses [1,2,3,4,5,6].

From 1930 to the present, thousands of viruses from 28 different families have been found in bats. [7]. Studies have shown that the number of viruses carried by bats is much greater than that carried by other orders of mammals [8].

Many of these viruses can cause severe and fatal illnesses in humans and other animals [9]. Some of these viruses were first recognized after they were associated with disease and death in humans or livestock, but most were isolated or found accidentally in general viral studies or during surveillance of specific viral pathogens. The global COVID-19 pandemic caused by the new coronavirus SARS-CoV-2 has increased the interest in bats as reservoirs of new and emerging viruses. Kazakhstan is located in the center of the Eurasian continent and is characterized by a large species, genetic, ecosystem and landscape biodiversity.

Currently, 27 species of bats belonging to 10 genera (*Rhinolophus*, *Myotis*, *Barbastella*, *Plecotus*, *Nyctalus*, *Pipistrellus*, *Eptesicus*, *Vespertilio*, *Otonycteris and Tadarida*) *from three families* (*Rhinolophidae*, *Vespertillionidae*, *Molossidae*) are registered on the territory of Kazakhstan. Several other species are known near its borders, most of which probably enter the country. This makes up more than 70% of the bat fauna of the former USSR [10]. In the composition of the bat fauna, two main faunal complexes are conditionally distinguished: the southern mountain-desert (16 species) and the boreal (9 species) complexes. The bat fauna of the southern part of Kazakhstan is richest in the mountains and foothills, where 80% of the species of the mountain–desert complex lives. Representatives of the boreal fauna are common in the forests of the north and east of the country. The southern and boreal faunas have little contact and penetrate each other. Five species of bats are listed in the Red Book of Kazakhstan and are protected by the state (*Myotis ikonnikovi*, *Barbastella leucomelas*, *Eptesicus bobrinskoi*, *Otonycteris hemprichii*, *Tadarida teniotis*).

To date, there have few little studies on bats in Kazakhstan, and little is known about the diversity and potential spread of bat-borne viruses.

In our previous study, we showed the presence of alphacoronaviruses in bats in Kazakhstan [11].

However, this research was limited to one area and focused on coronaviruses only. In this work, having increased the geography of the studied areas, we set the goal of studying the viral diversity in various populations of bats.

## 2. Materials and Methods

### 2.1. Bat Sampling

In the period from 2020 to 2022, 1149 samples (393 oropharyngeal swabs, 349 brain samples, 407 guano) were collected in 9 regions of Kazakhstan from 4 species of bats (*Vespertilio murinus*, *Nyctalus noctula*, *Myotis blythii*, *Eptesicus serotinus*). Detailed information about the sampling sites and collected samples are shown in Figure 1 and Figure 2.

Bat trapping and sampling were carried out in old, abandoned buildings, on rooftops, in an old tunnel, and in a cave. The Batbox Baton Bat Detector (NHBS, London, U.K.) ultrasonic detector series D 100 was used to locate the bats. Bats in tunnels or caves were caught with mist nets, and bats inhabiting abandoned buildings and rooftops were caught using anatomical tweezers. Guano and oropharyngeal swabs were collected in each colony, and several individuals were taken for brain sampling. Swabs were taken from all captured bats. Some of these animals were seized for brain sampling. To do this, we seized several (1–30) bats (not rare species) from each colony, depending on the size of the colony. On the roofs of houses and in abandoned buildings, one individual was seized from each colony, and up to 30 individuals were captured in tunnels or caves. The animals that were not seized for brain sampling were released after swab collection. The captured bats were placed in special cages and delivered alive to the laboratory.

Oropharyngeal swabs were collected from the bats using polyester swabs, which were placed in cryovials containing the transport medium (PBS with antibiotics). Clean medical gauze was laid out on the floor to collect guano under the bat colonies. After 3–4 h, guano was collected in separate test tubes from several areas of the gauze surface. The sample tubes were labeled and placed in a Dewar vessel with liquid nitrogen for transport to the laboratory. Upon arrival at the laboratory, all samples were stored at −40 °C until examination.

The study was approved by the ethics committee of the Research Institute for Biological Safety Problems (No. 5-13.06/20, 13 July 2020). The work was carried out under the conditions of BSL-3 laboratory.

Upon arrival at the laboratory, the live bats were anesthetized and euthanized by intravenous administration of potassium chloride combined with a 70% CO_2_ inhalant according to the guidance of Michigan Rabies Working Group [12].

They were quickly beheaded with sharp scissors, and their brains were removed from the skulls. The extracted brains were placed in phosphate-buffered saline (PBS) tubes and frozen at −40 °C. The frozen tissues were used to prepare a 10% suspension in PBS for RNA isolation.

### 2.2. RNA Extraction

The RNA was isolated with the QIAmp Viral RNA Mini Kit (Qiagen, Hilden, Germany) according to the manufacturer’s instructions.

### 2.3. PCR Study of the Samples from Bats

#### 2.3.1. PCR for Influenza Virus Type A

The following primers and probe were used to detect the influenza type A virus. Forward primer M + 25 AGA TGA GTC TTC TAA CCG AGG TCG, M − 124 TGC AAA AAC ATC TTC AAG TCT CTG, M + 64 FAM-TCA GGC CCC CTC AAA GCC GA –TAMRA [13]. A/chicken/Astana/6/2005 (H5N1) was used as a positive control. DNase-/RNase-free water was used as a negative control. The One-Step RT-PCR Kit (Qiagen, Hilden, Germany) was used for the work. PCR master mix components: 5 × buffer, 4 µL, MgCl_2_, 1 µL, dNTPs, 0.8 µL, 10 pmol concentrations of primer M + 25 and primer M − 124, as well as a 0.3 μM solution of probe M + 64, enzyme mix, 0.8 µL, RNA, 5 µL and water to a total of 20 μL. RT-PCR amplification program for the detection of influenza type A: 50 °C, 30 min; 94 °C, 15 min (reverse transcription); 35 amplification cycles: 94 °C, 00 s, 60 °C, 20 s.

#### 2.3.2. RT-PCR for Detecting Coronaviruses

The following primers were used to detect the coronaviruses. Forward primer PanCor IN-6: GGT TGG GAC TAT CCT AAG TGT GA and reverse primer PanCor IN-7: CCA TCA TCA GAT AGA ATC ATC ATA [11]. Plasmid RNA containing an insert corresponding to the RNA-dependent reverse polymerase (RdRp) gene was used as a positive control. DNase-/RNase-free water was used as a negative control. The SuperScript™ III One-Step RT-PCR System with Platinum™ Taq DNA Polymerase reaction kit (Invitrogen, Carlsbad, CA, USA) was used for the work. PCR master mix components: 2 × buffer, 12.5 µL, PanCor IN-6 forward primer, 1 µM, PanCor IN-7 reverse primer, 1 µM, SSCIII/Tag Enzyme mix, 1 µL, RNA, 2 µL and water to a total of 25 μL. the total volume of the PCR master mix was 25 µL. RT-PCR amplification program for the detection of coronaviruses: 95 °C, 10 min; 95 °C, 1 min (reverse transcription); 40 amplification cycles: 95 °C, 1 min, 56 °C–1 min, 72 °C, 1 min; 72 °C, 5 min.

#### 2.3.3. Real-Time RT-PCR for Detecting the Middle East Respiratory Syndrome Coronavirus (MERS-CoV)

The following primers and probe were used to detect the UpE gene of the MERS-CoV virus: upE-Fwd forward primer GCAACGCGCGATTCAGTT, upE-Rev reverse primer GCCTCTACACGGGACCCATA, and upE-Prb [FAM]-CTCTTCACATAATCGCCCCGAGCTCG-TAMRA probe [14]. Plasmid RNA (Institute of Virology, New York, NY, USA) containing an insert corresponding to the E gene was used as a positive control. DNase-/RNas-free water was used as a negative control. The SuperScript III One-Step RT-PCR system with Platinum Taq DNA polymerase kit (Invitrogen, Carlsbad, CA, USA) was used for the work. PCR master mix components: 2 × buffer, 12.5 µL, MgSO_4_, 0.4 µL, 400 nM concentrations of primer upE-Fwd and primer upE-Rev, as well as a 200 nM solution of probe upE-Prb, SSCIII/Tag enzyme mix, 1 µL, RNA, 5 µL and water to a total of 25 μL. RT-PCR amplification program for the detection of the MERS-CoV virus UpE gene: 55 °C, 30 min; 94 °C, 15 min (reverse transcription); 45 amplification cycles: 94 °C, 15 s, 58 °C, 30 s.

#### 2.3.4. Real-Time RT-PCR Confirmatory Assay (MERS-CoV)

The following primers and probe were used to detect the ORF1b gene (open reading frame 1b gene) of the MERS-CoV. Forward primer ORF1b-Fwd TTCGATGTTGAGGGTGCTCAT, reverse primer ORF1b-Rev TCACACCAGTTGAAAATCCTAATTG, probe ORF1b-Prb [FAM]-CTCTTCACATAATCGCCCCGAGCTCG-TAMRA [14]. Plasmid RNA (Institute of Virology, New York, NY, USA) containing an insert corresponding to the ORF1b gene was used as a positive control. DNas-/RNase free water was used as a negative control. The SuperScript III One-Step RT-PCR system with the Platinum Taq DNA polymerase kit (Invitrogen, Carlsbad, CA, USA) was used for the work. PCR master mix components: 2 × buffer, 12.5 µL, MgSO_4_, 0.4 µL, 400 nM concentrations of primer ORF1b-Fwd (10 µM) and primer ORF1b-Rev, as well as a 200 nM solution of probe upE-Prb, SSCIII/Tag enzyme mix, 1 µL, RNA, 5 µL and water to a total of 25 μL. RT-PCR amplification program for the detection of the MERS-CoV ORF1b gene: 55 °C, 30 min; 94 °C, 15 min (reverse transcription); 45 amplification cycles: 94 °C, 15 s, 58 °C, 30 s.

#### 2.3.5. Real-Time RT-PCR for Detecting the Crimean Congo Hemorrhagic Fever (CCHF) Virus and Coxiella Burnetii DNA

For the detection of the RNA of the Crimean Congo hemorrhagic fever virus and the DNA of the causative agent of Q fever, a set of reagents for real-time PCR, “OM-Screen-CCHF/Q-RT” (Syntol, Russia), was used according to the manufacturer’s instructions. Components of the PCR master mix: reaction buffer (RB), 15 µL, sample/negative control/positive control, 20 µL. Kit amplification program: 50 °C, 15 min; 95 °C, 5 min (reverse transcription); 50 amplification cycles: 60 °C, 40 s (fluorescence signal reading), 95 °C, 15 s.

#### 2.3.6. Real-Time RT-PCR for Detecting West the Nile fever (WNF) Virus and Rift Valley Fever (RVF) Virus

For the detection of the West Nile fever virus and Rift Valley fever virus RNAs, the “OM-Screen-WNF/RVF-RT” (Syntol, Moscow, Russia) was used, according to the manufacturer’s instructions. Components of the PCR master mix: RB, 15 µL, sample/negative control/positive control, 20 µL. Kit amplification program: 50 °C, 15 min; 95 °C, 5 min (reverse transcription); 50 amplification cycles: 60 °C, 40 s (fluorescence signal reading), 95 °C, 15 s.

#### 2.3.7. Real-Time RT-PCR for Detecting the Tick-Borne Encephalitis (TBE) Virus

For the detection of the tick-borne encephalitis virus RNA, the “OM-Screen-TBE-RT” (Syntol, Russia) was used, according to the manufacturer’s instructions. Components of the PCR master mix: RB, 15 µL, sample/negative control/positive control, 20 µL. Kit amplification program: 50 °C, 15 min; 95 °C, 5 min (reverse transcription); 50 amplification cycles: 60 °C, 40 s (fluorescence signal reading), 95 °C, 15 s.

#### 2.3.8. Real-Time RT-PCR for Detecting Hantaviruses

For the detection of hantavirus RNA—the causative agents of hemorrhagic fever with renal syndrome (HFRS)—we used “OM-Screen-HFRS-RT” (Syntol, Russia) which allows determining each of the 4 species of the genus Hantavirus (Puumala, Dobrava, Hantaan, Seoul), according to the manufacturer’s instructions. Components of the PCR master mix: RB-15, µL, sample/negative control/positive control, 20 µL. Kit amplification program: 50 °C, 15 min; 95 °C, 5 min (reverse transcription); 50 amplification cycles: 95 °C, 15 s; 58 °C, 20 s (fluorescence signal reading), 72 °C, 20 s.

#### 2.3.9. Real-Time RT-PCR for Detecting the Rabies Virus

For the detection of the rabies virus RNA, a set of reagents for real-time PCR (Rabies virus) was used according to the manufacturer’s instructions (LLC Fractal Bio, Russia). We used 15 µL of the RT-PCR mix with 0.5 µL of the enzyme, took 15 µL of the mixture and combined it with 10 µL of the sample/positive control/negative control. Reverse transcription and PCR program: 37 °C, 15 min; 95 °C, 3 min (reverse transcription); 40 amplification cycles: 60 °C, 30 s, 95 °C, 10 s.

#### 2.3.10. RT-PCR for the Rabies Virus Nucleoprotein Gene Product

The PCR product of the rabies virus nucleoprotein gene was obtained using the SuperScript III One-Step RT-PCR System with the Platinum Taq DNA polymerase kit (Invitrogen, Carlsbad, CA, USA) in accordance with the manufacturer’s instructions. A pair of Rab-NF primers, ATGGATGCCGACAAGATTGT and Rab-NR-GCACACTGTTGTTCAACTCC, was used to amplify the PCR product of the rabies virus nucleoprotein gene. The size of the PCR product was 1372 bp [15]. The CVS strain of the rabies virus was used as a positive control. DNase-/RNas-free water was used as a negative control. PCR master mix components: H_2_O, 2.6 µL, 2 × buffer, 12.5 µL, MgSO_4_, 0.4 µL, forward primer Rab-NF (20 pmole), 1 µL, reverse primer Rab-NR (20 pmole), 1 µL, SSCIII/Tag enzyme mix, 1 µL, RNA, 5 µL, total volume of the PCR master mix, 25 µL. RT-PCR amplification program for generating the N-gene of the rabies virus: 50 °C, 30 min; 95 °C, 2 min (reverse transcription); 30 amplification cycles: 94 °C, 20 s, 50 °C, 20 s, 68 °C, 3 min; 68 °C, 10 min; 4 °C, ∞.

### 2.4. Sequencing Assays, BLASTn Analysis and Phylogenetic Analyses

For sequencing, the obtained PCR products were used as templates. Sequencing was performed on an Applied Biosystems 3130 automated DNA sequencer (Hitachi, Tokyo, Japan) using the BigDye Terminator v3.1 Cycle Sequencing kit (Applied Biosystems, Inc., Vilnius, Lithuania). The resulting nucleotide sequences were analyzed in Sequencer v. 4.5 (Gene Codes Corporation, Ann Arbor, MI, USA).

The identities and similarities of the sequenced isolates were analyzed using the Basic Local Alignment Search Tool (BLASTn) of the National Center for Biotechnology Infor-mation (NCBI) GenBank database.

The evolutionary history was inferred using the Neighbor-Joining method [16]. The evolutionary distances were computed using the Maximum Composite Likelihood method [17] and are in the units of the number of base substitutions per site. This analysis involved 60 nucleotide sequences. Evolutionary analyses were conducted in MEGA11 [18]

The analyses used available nucleotide sequences of strains in GenBank:

Alphacoronaviruses: OM470277, OM4704349, OM470234, OM470014, MG000866, MG000871, MN611517, OM470035, OM470057, MT586838, MT586843, MG963199, MG963190, KY073747, MH170125, MH170089, MH170090, JX174639, MG963189, MG963191, KT253274, KT253280, KT253275, OM470219, MZ293735, KY073746, KY073744, MN183169, HQ184169, HQ184050, MK603150, MK603151, MK603157, MK603155, MK603152, MK603153, MK603154, MK603156, MK603160, MK603159, MK603161, DQ648835, HQ184061, HQ184058, HQ184054, HQ184057, KF843854, KF843862, MK603158, KT345294, KT345295.

Lyssaviruses: KJ748633, KX533959, MG383886, AB571007, KJ958262, KX533960, AY352481, KY242672, KJ958228, AY352490, KJ958249, AB570997, KT965734, AY352491, JX987744, AY352472, KT965736, KT965735, KT965738, JQ944705, KP997032, KT965737, KY765901, KT965733, JQ944708, U43432, U42702, GU9922302, KX148164, KX148191, KX148186, KF155000, KX148209, KT119772, KF155001, U22643, KF155002, MN784133, MG383884, KT894578, KM366206, KM366267, KU244266, AF006497, MF960865, NC020808, NC025385, MF197740, U22846, NC055474, NC020809, OU524420, U22848, U22844, U22845, MZ501949, NC031955, JX193798, S59448, NC025365, U22842.

## 3. Results

### 3.1. Detection of RNA Viruses in Bats

In 2020–2022, 1149 samples from 4 species of bats were collected in 9 regions of Kazakhstan, of which 140 were collected in the Aktobe region (73 swabs/6 guano/61 brains), 3 in the Almaty region (3/0/0), 113 in the Atyrau region (62/4/47), 24 in the East Kazakhstan region (12/0/12), 40 in the Jambyl region (20/0/20), 218 in the West Kazakhstan region (77/78/63), 191 in the Kyzylorda region (40/111/40), 145 in the North Kazakhstan region (47/51/47), 275 in the Turkistan region (59/157/59). The samples were collected from *Vespertilio murinus* (*n* = 621) (54.1%) in the Jambyl, Turkistan, Kyzylorda, North Kazakhstan regions, from *Myotis blythii* (*n* = 19) (1.6%) in the Jambyl and Almaty regions, from *Nyctalus noctula* (*n* = 14) (1.2%) in the Turkistan region, and from *Eptesicus serotinus* (*n* = 495) (43.1%) in the Aktobe, Atyrau, East Kazakhstan, West Kazakhstan regions.

The brain samples were tested for the presence of the rabies virus RNA. Guano and swabs were tested for the presence of the RNA viruses SARS-CoV-2, MERS, TBE, CCHF, WNF, influenza A and alphacoronaviruses.

Alphacoronavirus RNA was found in 38 (4.75% of whole 800 guano and swabs) bat samples. The PCR positive samples were 33 (8.4%) of 393 swabs and 5 (1.2% of 407 guano) guano samples. Alphacoronaviruses were found in three bat species studied. The infection rates of the bats were: species Vespertilio murinus, 0.85% (470 tested/4 positives), Eptesicus serotinus, 9.94% (312/31), Myotis blythii, 27.27% (11/3). The prevalence of coronavirus among bats in the Almaty region was 100% (3 samples/3 positive samples), in the Atyrau region, it was 18.18% (66/12), in the Aktobe region, 13.92% (73/11), in the West Kazakhstan region, 5.16% (155/8), in the Turkistan region, 1.39% (216/3), in the Kyzylorda region, 0.66%(151/1) (Figure 3). In the East Kazakhstan, Jambyl and North Kazakhstan regions, coronavirus RNA was not found in samples from bats.

For the first time, it was established that bats are infected with the rabies virus in the territory of the Republic of Kazakhstan. The rabies virus RNA was detected in 27 (7.74%) brain samples from 349 bats. The prevalence of the rabies virus among bats in the North Kazakhstan region was 12.77% (47 samples/6 positive samples), in the Atyrau region, 17.02% (47/8), in the Aktobe region, 1.64% (61/1), in the West Kazakhstan region, 11.11% (63/7), in the Turkistan region, 5.08% (59/3), in the Jambyl region, 10.0% (20/2) (Figure 4). In the East Kazakhstan, Kyzylorda and Almaty regions, the RNA of the rabies virus was not detected in samples from bats. *Vespertilio murinus* had an infection rate of 7.28%, *Eptesicus serotinus* of 8.74%.

### 3.2. Coronaviruses Sequencing and Phylogenetic Analysis

We managed to obtain sequences of the RdRp gene (416−425 bp) from 12 bats samples collected in 5 regions of Kazakhstan (West Kazakhstan, Aktobe, Atyrau, Kyzylorda and Turkistan regions). A phylogenetic analysis made it possible to attribute all sequences to alphacoronaviruses and divide them into two different groups (Figure 5). Eleven sequences formed a separate new monophyletic single clade with 100% nucleotide sequence identity. This clade was phylogenetically similar to African sequences. The closest to this group were sequences obtained from bats in Ghana (D > 0.0693), Zimbabwe (D > 0.0737) and Kenya (D > 0.0685). One sequence (BatCov-47-2021) from a bat from the Turkistan region was markedly different (nucleotide identity: 71.98%) from the other 11 sequences and was identical to the Kazakh isolates previously isolated in the southern regions of Kazakhstan (D > 0.004).

Sequences of the RdRp gene from 12 bats samples. GenBank accession numbers: OP562947 (*Eptesicus serotinus*); OP605949 (*Eptesicus serotinus*); OP605948 (*Eptesicus serotinus*); OP562946 (*Eptesicus serotinus*); OP562945 (*Eptesicus serotinus*); OP562944 (*Eptesicus serotinus*); OP562943 (*Eptesicus serotinus*); OP562942 (*Eptesicus serotinus*); OP562941 (*Vespertilio murinus*); OP562940 (*Eptesicus serotinus*); OP562939 (*Eptesicus serotinus*); OP605947 (*Vespertilio murinus*).

### 3.3. Rabies Virus Sequencing and Phylogenetic Analysis

From the 27 PCR positive samples, we were able to obtain one sequence of the nucleoprotein gene (1023 bp) of the rabies virus from *Eptesicus serotinus*. A phylogenetic analysis of the N gene sequence OP585396 KZ(West)/bat/111/2021 showed a close relationship with Cosmopolitan (Central Asia) rabies group isolates (Figure 6). The genetic distance of the sequence OP585396 KZ(West)/bat/111/2021 from the sequence KT965737 KZ(West)/cattle/5328 (KT965737) was D > 0.00491, and that from with the sequence Russia/PO-014_2014_Primye (KP9932) was D > 0.00393.

## 4. Discussion

It is known that bats constitute one of the largest groups of mammalian species and are reservoirs for many viruses that can cause severe diseases in humans and animals [7]. However, little is known about the diversity and potential distribution of bat-borne viruses in Kazakhstan. This study is one of the few works devoted to the study of bats inhabiting Kazakhstan and of the various pathogens they may carry.

This paper presents the results of a study of 1149 samples from 4 species of bats collected in 9 regions of Kazakhstan. In our study, new alphacoronaviruses as well as the rabies virus were found circulating among bats in various regions of Kazakhstan.

Of the four known genera of coronaviruses, two genera (alphacoronaviruses and betacoronaviruses) are of public health importance [19], and some of them are of bat origin. It is known that bats from the families *Rhinolophidae*, *Hippoideridae*, *Emballonuridae*, *Miniopteridae* and *Vespertilionidae* are the reservoir of various bat coronaviruses in Asia [20]. However, there is little information on the diversity of bat coronaviruses in the regions of Central Asia. The only study we previously conducted showed that *Myotis blythii* bats from the southern regions of Kazakhstan are carriers of alphacoronaviruses. In three different colonies of the Turkistan region, two different lines of bat alphacoronaviruses were found, which were closely related to bat coronaviruses from China, France, Spain and South Africa [11].

In this study, we expanded the geography of the search. The study was carried out in bat colonies inhabiting nine regions of Kazakhstan. An examination of bat samples by PCR showed that 4.7% were positive for coronaviruses. The positive samples were collected from bats *Myotis blythii* in Almaty, *Eptesicus serotinus* in the Aktobe, Atyrau and West Kazakhstan regions, *Vespertilio murinus* in the Turkistan and Kyzylorda regions. A phylogenetic analysis of the RdRp gene sequences of bats from different regions of Kazakhstan made it possible to classify them as alphacoronaviruses and divide them into two groups. One sequence (BatCov-47-2021) was close to those previously identified in southern Kazakhstan. This was to be expected, as this sample was collected in the Kepterhan tunnel in the Turkistan region, where we isolated similar viruses.

The main new group (*n* = 11) differed significantly from previously obtained sequences and was closely related to bat coronaviruses from Ghana (D > 0.0693), Zimbabwe (D > 0.0737) and Kenya (D > 0.0685). This group consisted of sequences from samples collected in four regions (West Kazakhstan, Aktobe, Atyrau and Kyzylorda) from two species of bats (*Eptesicus serotinus*, *Vespertilio murinus*). Despite the geographical remoteness, the sequences of this group were 100% identical to each other. Thus, the data obtained show that the various alphacoronaviruses circulate among bats in the southern and western regions of Kazakhstan. The data suggest that the co-circulation of coronaviruses occurs in several bat species with overlapping geographic distributions. To clarify the obtained data, it is necessary to continue an active surveillance with the inclusion of whole-genome sequencing in the bat populations of Kazakhstan.

Bats are a reservoir of lyssaviruses, including the rabies virus. Four different types of lyssaviruses are known to circulate in bats in Europe [21]. To date, not a single lyssavirus has been isolated from bats in Kazakhstan, although rabies is widespread in Kazakhstan. Natural foci of rabies exist in almost all areas, and animal diseases are recorded annually [22]. In our study, for the first time, the infection of bats with the rabies virus was established in the territory of the North Kazakhstan, Atyrau, Aktobe, West Kazakhstan, Turkistan and Jambyl regions. A total of 27 (7.74%) PCR positive samples were obtained from two bat species (*Vespertilio murinus*, *Eptesicus serotinus*). However, it is necessary to interpret these data with caution, since the rabies virus has not been isolated from bats in the Eurasian continent before. Humans have been known to become infected after contact with bats. The available literature contains information about eight cases of human death from rabies after being bitten by bats from 1954 to 2007, of which two cases each in Ukraine and Russia, and one case each in Finland, Great Britain, China and India. Only in four of these cases, the type of virus was determined (EBLV-1, EBLV-2, Irkut) [23]. In addition, in the available literature, we could not find data on the sensitivity and specificity of the PCR test we used. We were able to obtain the rabies virus sequence from one bat. A phylogenetic analysis of the nucleotide sequence of the N gene of the rabies virus obtained from a sample of *Eptesicus serotinus* made it possible to assign it to the I genetic cluster of the Cosmopolitan clade, subclade Central Asia. These data are consistent with previously obtained data and confirm that rabies viruses in Kazakhstan are closely related to viruses from Asia and Europe [15].

Thus, the obtained data show that bats inhabiting the territory of Kazakhstan are carriers of various alphacoronaviruses and rabies virus. These data provide new information and increase our knowledge about the distribution of pathogens in bat populations. However, these data are insufficient to understand the role of bats in the epidemiology of various infectious diseases. Therefore, it is necessary to continue research in bat populations to improve the surveillance of infectious diseases.

## Figures and Tables

**Figure 1 viruses-14-02743-f001:**
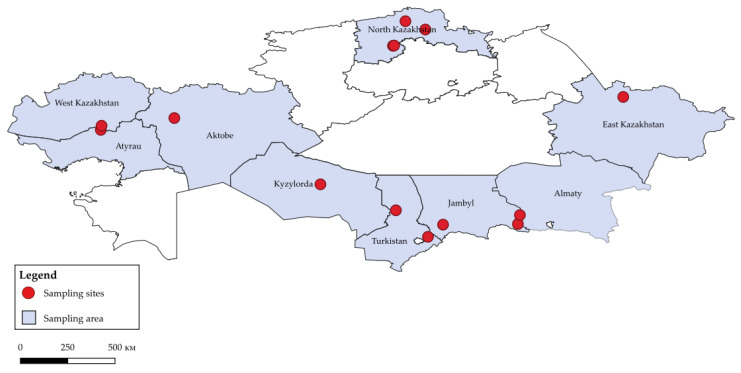
Map of Kazakhstan with bats sampling sites.

**Figure 2 viruses-14-02743-f002:**
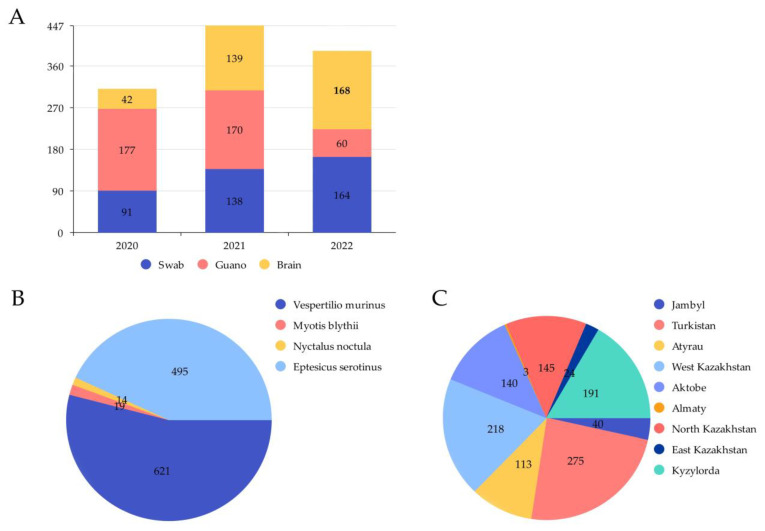
Characterization of the collected samples. (**A**) Graph showing the distribution of the number of samples taken from 2020 to 2022. (**B**) Number of samples taken by region. (**C**) Number of samples taken from different species of bats.

**Figure 3 viruses-14-02743-f003:**
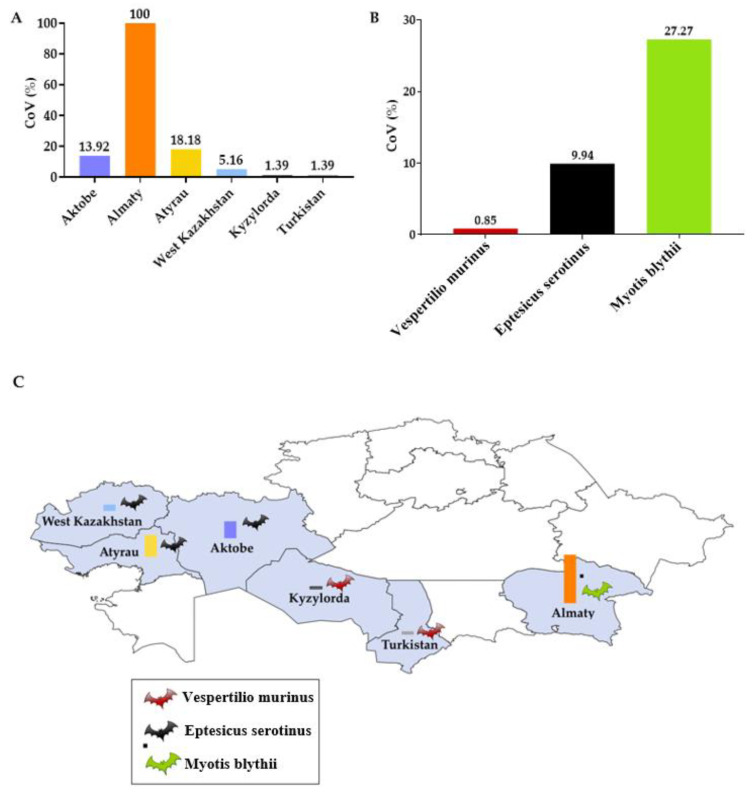
Alphacoronavirus prevalence among bats in Kazakhstan. (**A**) by regions; (**B**) by species of bats; (**C**) distribution map.

**Figure 4 viruses-14-02743-f004:**
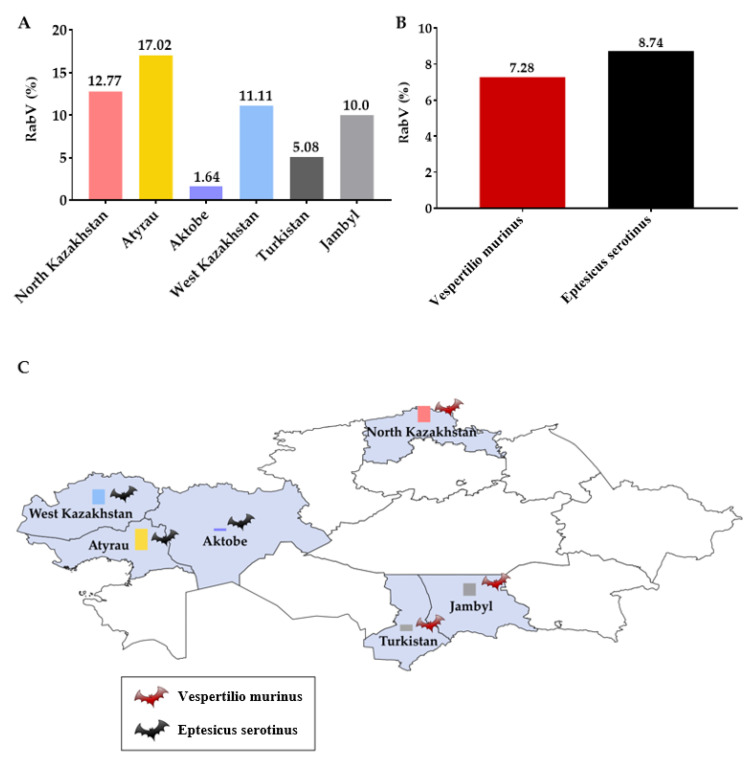
Prevalence of the rabies virus among bats in Kazakhstan. (**A**) by regions; (**B**) by species of bats; (**C**) distribution map. The RNAs of SARS-CoV-2, MERS, TBE, CCHF, WNF, influenza A viruses were not detected in the bat samples.

**Figure 5 viruses-14-02743-f005:**
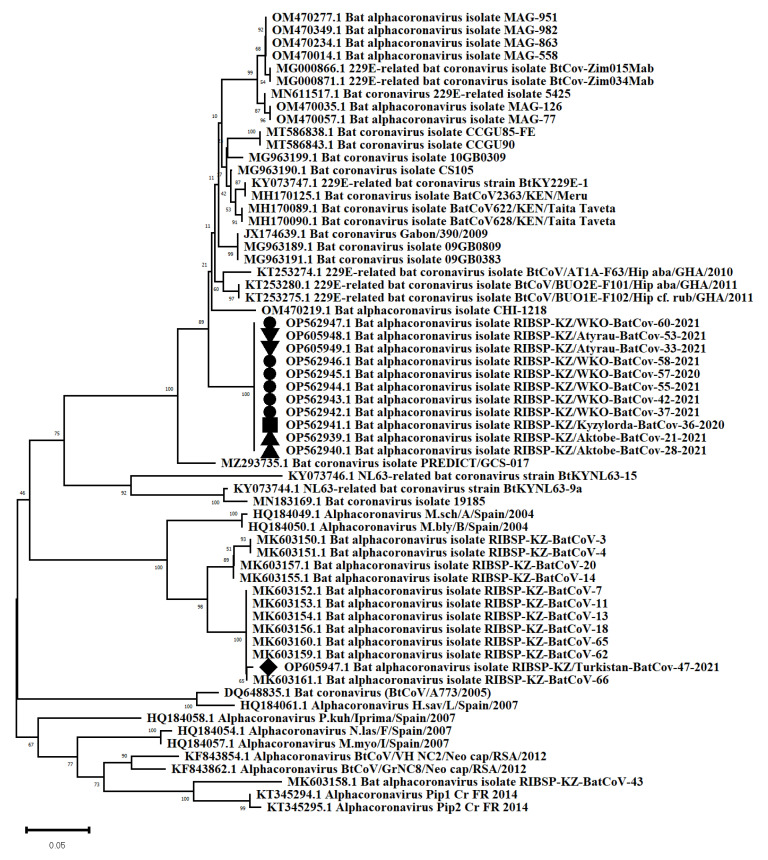
Phylogenetic analysis of the nucleotide gene sequence of RNA-dependent reverse polymerase (RdRp).

**Figure 6 viruses-14-02743-f006:**
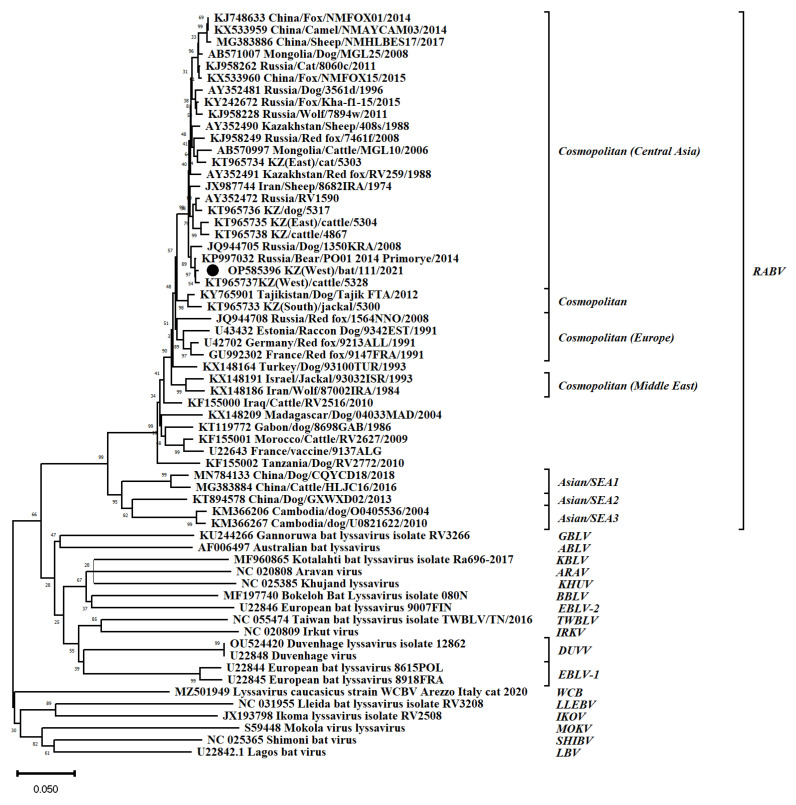
Phylogenetic analysis of nucleotide sequences of the nucleoprotein gene (N-gene) of the rabies virus.

## Data Availability

The obtained gene fragments of viruses were deposited in GenBank under accession numbers OP562947, OP605949, OP605948, OP562946, OP562945, OP562944, OP562943, OP562942, OP562941, OP562940, OP562939, OP605947, OP585396.

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
