# Peer review of "The Prevalence of Viral Pathogens among Bats in Kazakhstan"

_viruses, 2022, doi:10.3390/v14122743_

Round 1
Reviewer 1 Report
This manuscript titled “The prevalence of viral pathogens among bats in Kazakhstan” focuses on a current research need for the region (Central Asia) and Kazakhstan, especially due to the limited data available on this topic.
Further in view of spread of different pathogens during different time periods, the role of bats as reservoirs is a critical issue.
This study has investigated the prevalence of a set of viruses (which can cause severe disease in humans) in four species of bats in 9 different regions of Kazakhstan and report the presence of alphacoronaviruses in three species and rabies virus in two bat species as well as the phylogeny of the viruses.
Following are some clarifications which needs to be addressed,
1. The previous study was conducted in Turkistan in the Southern region of Kazakhstan. Of the 14 different regions in Kazakhstan, what were the criteria to select the 9 regions mentioned in this study ?
2. Of the 27 species of bats found in Kazakhstan, are the four species mentioned in the study the most common species and/or were they the most commonly found in the collection sites ?
3. Were the brain samples PCR negative ?
4. Line 256 mention “Alphacoronaviruses were found in all four bat species studied.”
Although this sentence says positive in all four species, data (figure 3) and text show data for only three species ?
5. Figure 3 B – need to mention how the infection rate was calculated.
Other comments
Introduction could include any information available on the common bat species in Kazakhstan.
Also need to mention if any previous studies are available on bat pathogens in Kazakhstan, other than the already mentioned info by the authors.
Minor comments
Some of the editorial changes that needs to be addressed are mentioned in the text.
Author Response
Response to Reviewer 1 Comments
Dear Editor,
Thank you for your positive response to our article that was recently submitted to Viruses:
The prevalence of viral pathogens among bats in Kazakhstan.
Below is a detailed response to the suggestions.
We look forward to hearing from you.
Kind Regards,
Takhmina Argimbayeva
Explanations to consider
Point 1: The previous study was conducted in Turkistan in the Southern region of Kazakhstan. Of the 14 different regions in Kazakhstan, what were the criteria to select the 9 regions mentioned in this study ?
Response 1: The territory of Kazakhstan is huge and occupies from the eastern outskirts of the Volga delta in the west to the Altai Mountains in the east, from the West Siberian Plain in the north to the Tien Shan mountain system in the south of the country. The total area is 2724.9 thousand km2. It was practically impossible to explore the entire territory of Kazakhstan within the framework of one small project (youth project), since funds were limited. And collecting samples is very expensive. Initially, we planned to sample annually only four regions in the north, west, south and east (one point for each region). But later, as part of the project, we had the opportunity to collect more samples. Thus, another project was additionally financed in our institute, within the framework of which it was planned to collect samples from farm animals in various regions of Kazakhstan. As part of this project, we additionally collected samples from bats in other regions. Therefore, samples were collected in 9 regions, and not 4 regions as planned.
Point 2: Of the 27 species of bats found in Kazakhstan, are the four species mentioned in the study the most common species and/or were they the most commonly found in the collection sites ?
Response 2: 27 species live in Kazakhstan, of which 5 species of bats are listed in the Red Book of Kazakhstan and they are protected by the state (Myotis ikonnikovi, Barbastella leucomelas, Eptesicus bobrinskoi, Otonycteris hemprichii, Tadarida teniotis). The four species from which we have sampled are the most common and frequent species in these regions. In addition, they were most common in regional sampling.
Point 3: Were the brain samples PCR negative ?
Response 3: We collected 349 brain samples and only 27 were positive for PCR. The rest were negative for PCR.
Point 4: Line 256 mention “Alphacoronaviruses were found in all four bat species studied.”
Although this sentence says positive in all four species, data (figure 3) and text show data for only three species ?
Response 4: Thanks for the comment. We have made changes to the text. Alphacoronaviruses have been found in three of the four bat species studied.
Point 5: Figure 3 B – need to mention how the infection rate was calculated.
Response 5: Done. We put it in the text.
Other comments
Point 6: Introduction could include any information available on the common bat species in Kazakhstan.
Response 6: Done. We put it in the text.
Point 7: Also need to mention if any previous studies are available on bat pathogens in Kazakhstan, other than the already mentioned info by the authors.
Response 7: Previously, studies on the carriage of pathogens by bats in Kazakhstan have not been conducted. The first study was conducted by us in 2016-2018. The results are published in the article: Mendenhall I. H., Kerimbayev A.A., Strochkov V.M., Sultankulova K.T., Kopeyev S.K., Su Yvonne CF, Smith G JD, Orynbayev M.B. Discovery and characterization of novel bat coronavirus lineages from Kazakhstan. Viruses 2019, 11, 356; doi:10.3390/v11040356.
Minor comments
Point 8: Some of the editorial changes that needs to be addressed are mentioned in the text.
Response 8: Did not find comments on the text

Reviewer 2 Report
This is an interesting study, using an impressive number of samples to to test for a range of viruses in poorly-studied bat populations throughout Kazakhstan. However, I have concerns about the claims of widespread circulation of rabies virus in bats. Rabies virus has never been found in bats outside of the Americas, so the detection of such circulation in Eurasian bats would be very concerning to public health agencies if true. As such, these claims have to be backed up by very strong evidence, which is not currently the case. At least one of the species implicated (Eptesicus serotinus) is known to be infected by European Bat Lyssavirus 1 (PMID: 30747140), a virus in the same genus as rabies virus but without the same level of public health concern. There therefore needs to be evidence that the sequence in question is truly rabies virus, and not some other lyssavirus, and that the PCR tests used are specific enough to conclude that it is rabies virus that was detected in all unsequenced samples. Indeed, given the frequency with which novel lyssavirus species have been detected in the area around Kazakhstan (see fig. 2 of PMID: 30747140), even a new species seems more likely to me than a host shift of canine rabies towards Eurasian bats. I have some specific suggestions for how the data analysis could be improved to resolve this below.
Major points:
Line 65: No table 1 was included in the review documents.
Line 70: How can it be catch-release if bats were euthanized? If it is the case that different locations or sample types were sampled differently, perhaps it would help to separate out the different sampling strategies into distinct paragraphs so all the related information is together and easier to follow. Related to this, were swab samples taken from the same bats which were also euthanised for brain samples, or are these different individuals? Were all the different sample types collected at the same sampling locations?
Line 194 - 214: Please give more details about the kit used (at the very least, it's full name, but ideally also primer sequences if available). It would also be helpful to know if the manufacturer makes any mention of ability to detect other lyssaviruses - is it definetely a rabies virus-specific kit?
Line 226 – 228: This phylogenetic analysis makes a lot of assumptions about the underlying sequences, which might explain why the placement of the new sequences generated here is so uncertain (low bootstrap support, see below). It would be better to do a search for the best-fitting DNA substitution model and then construct a maximum likelihood phylogeny. Both of these steps can be performed in the MEGA 11 software used already.
Figure 5: Bootstrap support for the placement of the branch leading to the majority of detected coronavirus sequences is very low (27%). How were the sequences included for comparison chosen? Perhaps it would help to include a wider diversity of sequences. The phylogeny also needs to include an outgroup, at least when plotted like this.
Figure 6: As mentioned above, the conclusion that this is rabies virus needs to be supported by strong evidence. BLAST searches are mentioned in the methods section, but this is never discussed. Please include a phylogeny showing this sequence in the context of other lyssaviruses, and ideally also add an outgroup to this phylogeny. Without this, we have no way of knowing if the current placement is accurate (but the poor bootstrap support suggests it is not). Also, it is again unclear how the sequences included for comparison were chosen, as not all known rabies virus lineages are represented. Finally, please include information on the length of the sequence obtained (and also for coronaviruses earlier on).
Line 352 – 364: Even if the sequenced sample really does contain rabies virus, it would be unclear whether all of the detected PCR positives are the same species. Information about the specificity of the PCR test would help strengthen this conclusion. However, what is known about other lyssaviruses in bats (and particularly in the species detected as positive) should also be mentioned as a possible explanation.
Minor points:
Species names should be in italics.
Line 24: Many of the acronyms listed would not be known to a wide audience, so it would be better to write these out (SARS-CoV-2 is common enough, but what e.g. TBE and WNF are is unclear). It would also help to explain why these specific viruses are highlighted in this way – currently it only becomes clear in the methods section that tests were done for these specific viruses.
Line 51: It is unclear what "these 30 forms" refers to. The previous sentence discussed only 27 species.
Line 71: I think a count is missing here – numbers are given for abandoned buildings and tunnel/cave, but not for the first part of the sentence (roofs of houses).
Line 139: Please write the full virus name, not just the syndrome.
Line 257 – 260: Please include sample size alongside percentages.
Data availability:
I was unable to verify the data availability statement, but I assume sequences will be released upon publication.
Author Response
Response to Reviewer 2 Comments
Dear Editor,
Thank you for your positive response to our article that was recently submitted to Viruses:
The prevalence of viral pathogens among bats in Kazakhstan.
Below is a detailed response to the suggestions.
We look forward to hearing from you.
Kind Regards,
Takhmina Argimbayeva
Major points:
Point 1: Line 65: No table 1 was included in the review documents.
Response 1: Instead of a table, we have a chart (Figure 1) where all the data is included. Therefore, initially we excluded the table from the main text, and forgot to remove the word “table 1” from the text. We corrected the text and removed the word “table 1”.
Point 2: Line 70: How can it be catch-release if bats were euthanized? If it is the case that different locations or sample types were sampled differently, perhaps it would help to separate out the different sampling strategies into distinct paragraphs so all the related information is together and easier to follow. Related to this, were swab samples taken from the same bats which were also euthanised for brain samples, or are these different individuals? Were all the different sample types collected at the same sampling locations?
Response 2: Bats were captured for the collection of oropharyngeal swabs; after selection, they were released in the same place. However, due to the need for organ analysis, some captured bats from each locality were taken to the laboratory for research. We also took swabs from those bats that were taken for organ harvesting. The number of bats for research varied depending on the size of the colonies. We tried to do the least harm to the population of the colonies.
Point 3: Line 194 - 214: Please give more details about the kit used (at the very least, it's full name, but ideally also primer sequences if available). It would also be helpful to know if the manufacturer makes any mention of ability to detect other lyssaviruses - is it definetely a rabies virus-specific kit?
Response 3: Rabies virus RNA was detected using the "Rabies virus RNA detection kit" manufactured by LLC Fractal Bio, Russia. Unfortunately, manufacturers do not indicate either primer sequences or the possibility of detecting other lyssaviruses.
Point 4: Line 226 – 228: This phylogenetic analysis makes a lot of assumptions about the underlying sequences, which might explain why the placement of the new sequences generated here is so uncertain (low bootstrap support, see below). It would be better to do a search for the best-fitting DNA substitution model and then construct a maximum likelihood phylogeny. Both of these steps can be performed in the MEGA 11 software used already.
Response 4: Thank you. Done.
Point 5: Figure 5: Bootstrap support for the placement of the branch leading to the majority of detected coronavirus sequences is very low (27%). How were the sequences included for comparison chosen? Perhaps it would help to include a wider diversity of sequences. The phylogeny also needs to include an outgroup, at least when plotted like this.
Response 5: Thank you. Done.
Point 6: Figure 6: As mentioned above, the conclusion that this is rabies virus needs to be supported by strong evidence. BLAST searches are mentioned in the methods section, but this is never discussed. Please include a phylogeny showing this sequence in the context of other lyssaviruses, and ideally also add an outgroup to this phylogeny. Without this, we have no way of knowing if the current placement is accurate (but the poor bootstrap support suggests it is not). Also, it is again unclear how the sequences included for comparison were chosen, as not all known rabies virus lineages are represented. Finally, please include information on the length of the sequence obtained (and also for coronaviruses earlier on).
Response 6: Thank you. Done.
Point 7: Line 352 – 364: Even if the sequenced sample really does contain rabies virus, it would be unclear whether all of the detected PCR positives are the same species. Information about the specificity of the PCR test would help strengthen this conclusion. However, what is known about other lyssaviruses in bats (and particularly in the species detected as positive) should also be mentioned as a possible explanation.
Response 7: As we wrote earlier, the specificity and sensitivity of the PCR test used is unknown, so we cannot say that all of them are clearly positive for rabies. However, we can state with certainty that one sample from the Eptesicus serotinus bat is positive for the rabies virus. Phylogenetic sequence analysis obtained from this 1023 bp specimen suggests that it is a rabies virus. This sequence was identical to the rabies virus sequences and differed from other lisaviruses (see Figure 6).
We have made corrections to the text of the article.
Minor points:
Point 8: Species names should be in italics.
Response 8: Thank you. Done.
Point 9: Line 24: Many of the acronyms listed would not be known to a wide audience, so it would be better to write these out (SARS-CoV-2 is common enough, but what e.g. TBE and WNF are is unclear). It would also help to explain why these specific viruses are highlighted in this way – currently it only becomes clear in the methods section that tests were done for these specific viruses.
Response 9: This has been corrected. We wrote the full title at the first mention and in parentheses wrote an abbreviated title. And further on the text wrote only a abbreviated title. We hope that this will be clear to the readers.
Point 10: Line 51: It is unclear what "these 30 forms" refers to. The previous sentence discussed only 27 species.
Response 10: Corrected.
Point 11: Line 71: I think a count is missing here – numbers are given for abandoned buildings and tunnel/cave, but not for the first part of the sentence (roofs of houses).
Response 11: This has been corrected. One individual was caught on the roofs of houses and in abandoned buildings.
Point 12: Line 139: Please write the full virus name, not just the syndrome.
Response 12: Done.
Point 13: Line 257 – 260: Please include sample size alongside percentages.
Response 13: Done.
Data availability:
Point 14: I was unable to verify the data availability statement, but I assume sequences will be released upon publication.
Response 14: We were sent a number about data availability. But why they are still inaccessible to us is unknown. Probably will be published after the publication of the article.

Reviewer 3 Report
The authors conducted biosurveillance on bats from nine regions around Kazakhstan. Three different sample types (Oropharyngeal swab, feces and brain) were collected for various pathogen screening. Alphacoronaviruses were detected in 4.7% of samples tested, from all four bat species that were surveyed with varying prevalence per species. Bats sampled from 6/9 regions were positive for Alphacoronavirus. Rabies was detected in 7.74% of samples tested, from two bat species, in 6/9 regions. This is one of few studies conducting country wide zoonotic disease surveillance in Central Asia and will contribute greatly to the scientific community in understanding bat-borne virus diversity and distribution within the region.
The article will require revision to improve readability to the audience, with a focus on the methods description (refer to specific comments). This study appears to be focused on bat-borne viruses, however the list of viruses in the methods have a focus on livestock viruses (E.g. CCHF, TBEV). In addition, many viruses were included in the screening panel, but there were no further mention in the results or discussion.
Specific comments:
Figures: Figures and figure captions can be worked on slightly to improve visualization and communication to readers.
Figure 2. Figure 2A do not contribute significantly as the quantities can be visualized in Figure 2B.
Methods: Methods can be improved significantly.
Section 2.2. Header suggests only RNA extraction but include description of PCR methods.
Section 2.3. Header states “Confirmation of the presence of bat-borne pathogens by PCR method”. This would require rephrasing as most of the pathogens that were screened for are not bat-borne pathogens, but mostly livestock viruses.
Individual headings for each PCR section will require clarification or rephrasing. Some subheadings states PCR when it is a Real-time PCR instead (e.g. Influenza, MERS-CoV). Please check through Real-time PCR and/or RT-PCR (Reverse Transcription PCR).
Section 2.3.5 to 2.3.9. Please clarify if commercial kits were used for these assays. If so, please include kit and vendor name.
Section 2.3.5. Is Q fever part of this commercial kit? If so, please adjust section heading accordingly.
Section 2.3.6. As above. Combined commercial kit for WNFV and RVFV?
Section 2.3.7. As above.
Section 2.3.9. As above.
Were all the different sample types (E.g. brain, oropharyngeal swabs, feces) used in the different viruses screening? Or were only specific samples screened for specific viruses (E.g., rabies screening in brain samples only). This can be elaborated in the methods or results section.
Results:
Authors did not present data for any of the other viruses that were screened for in this study (E.g. CCHF, WNV). If samples were negative for these viruses, it would be worth a mention in the results or discussion.
Figure 3 and 4. Standardize the axis font sizes for easier visualization.
Figure 4. Caption mentioned SARS-CoV-2. Do the authors mean Coronavirus? Please rephrase caption for clarity.
Line 266. Authors did not elaborate on what samples were positive for Rabies virus.
Line 283. Do the authors mean Figure 5?
Coronavirus sequencing and Figure 5. Useful to include bat species where the cov sequences were obtained from.
Line 299. Similarly, useful to include bat species whereby rabies sequence was obtained from.
Discussion:
No mention of species, whereby the coronavirus sequences were obtained from.
Line 341 paragraph. Do these two bat species co-roosts, and therefore share the same viruses?
Author Response
Response to Reviewer 3 Comments
Dear Editor,
Thank you for your positive response to our article that was recently submitted to Viruses:
The prevalence of viral pathogens among bats in Kazakhstan.
Below is a detailed response to the suggestions.
We look forward to hearing from you.
Kind Regards,
Takhmina Argimbayeva
Specific comments:
Figures: Figures and figure captions can be worked on slightly to improve visualization and communication to readers.
Point 1: Figure 2. Figure 2A do not contribute significantly as the quantities can be visualized in Figure 2B.
Response 1: Done. We deleted Figure 2A.
Point 2: Methods: Methods can be improved significantly.
Response 2: Done.
Point 3:: Section 2.2. Header suggests only RNA extraction but include description of PCR methods.
Response 3: Corrected.
Point 4: Section 2.3. Header states “Confirmation of the presence of bat-borne pathogens by PCR method”. This would require rephrasing as most of the pathogens that were screened for are not bat-borne pathogens, but mostly livestock viruses.
Response 4: Corrected.
Point 5: Individual headings for each PCR section will require clarification or rephrasing. Some subheadings states PCR when it is a Real-time PCR instead (e.g. Influenza, MERS-CoV). Please check through Real-time PCR and/or RT-PCR (Reverse Transcription PCR).
Response 5: Corrected.
Point 6: Section 2.3.5 to 2.3.9. Please clarify if commercial kits were used for these assays. If so, please include kit and vendor name.
Response 6: Done.
Point 7: Section 2.3.5. Is Q fever part of this commercial kit? If so, please adjust section heading accordingly.
Response 7: Corrected.
Point 8: Section 2.3.6. As above. Combined commercial kit for WNFV and RVFV?
Response 8: Corrected.
Point 9: Section 2.3.7. As above.
Response 9: Corrected.
Point 10: Section 2.3.9. As above.
Response 10: Corrected.
Point 11: Were all the different sample types (E.g. brain, oropharyngeal swabs, feces) used in the different viruses screening? Or were only specific samples screened for specific viruses (E.g., rabies screening in brain samples only). This can be elaborated in the methods or results section.
Response 11: Done.
Results:
Point 12: Authors did not present data for any of the other viruses that were screened for in this study (E.g. CCHF, WNV). If samples were negative for these viruses, it would be worth a mention in the results or discussion.
Response 12: This was noted in the results after Figure 4. But for some reason this sentence (RNA of SARS-CoV-2, MERS, TBE, CCHF, WNF, influenza A viruses was not detected in bat samples) ended up in the title of Figure 4. Most likely this happened when the text was translated into the MDPI format. This has been corrected.
Point 13: Figure 3 and 4. Standardize the axis font sizes for easier visualization.
Response 13: Corrected.
Point 14: Figure 4. Caption mentioned SARS-CoV-2. Do the authors mean Coronavirus? Please rephrase caption for clarity.
Response 14: This sentence has nothing to do with this figure, see answer to comment 12). We have moved this sentence into the main text.
Point 15: Line 266. Authors did not elaborate on what samples were positive for Rabies virus.
Response 15: Only brain samples were tested for rabies virus. Therefore, the brain samples were positive. This is what we added to the text
Point 16: Line 283. Do the authors mean Figure 5? Coronavirus sequencing and Figure 5. Useful to include bat species where the cov sequences were obtained from.
Response 16: Thank you. Corrected. And on the text added the types of bats obtained sequences.
Point 17: Line 299. Similarly, useful to include bat species whereby rabies sequence was obtained from.
Response 17: Thank you. Done.
Discussion:
Point 18: No mention of species, whereby the coronavirus sequences were obtained from.
Response 18: Done.
Point 19: Line 341 paragraph. Do these two bat species co-roosts, and therefore share the same viruses?
Response 19: No. They don’t live together. Positive samples were collected from two species of bats in four regions of Kazakhstan.
The text reads as follows: This group includes sequences of samples collected in four regions (West Kazakhstan, Aktobe, Atyrau and Kyzylorda) from two types of bats. Despite the geographical distance, the sequences of this group were 100% identical to each other.
